# Primary Aldosteronism and Resistant Hypertension: A Pathophysiological Insight

**DOI:** 10.3390/ijms23094803

**Published:** 2022-04-27

**Authors:** Fabio Bioletto, Martina Bollati, Chiara Lopez, Stefano Arata, Matteo Procopio, Federico Ponzetto, Ezio Ghigo, Mauro Maccario, Mirko Parasiliti-Caprino

**Affiliations:** Endocrinology, Diabetes and Metabolism, Department of Medical Sciences, University of Turin, Corso Dogliotti 14, 10126 Turin, Italy; fabio.bioletto@unito.it (F.B.); bollati.martina@gmail.com (M.B.); chiara.lopez@fastwebnet.it (C.L.); stefano16196@gmail.com (S.A.); mprocopio35@gmail.com (M.P.); federico.ponzetto@unito.it (F.P.); ezio.ghigo@unito.it (E.G.); mauro.maccario@unito.it (M.M.)

**Keywords:** aldosterone, primary aldosteronism, arterial hypertension, resistant hypertension, secondary hypertension, pathophysiology

## Abstract

Primary aldosteronism (PA) is a pathological condition characterized by an excessive aldosterone secretion; once thought to be rare, PA is now recognized as the most common cause of secondary hypertension. Its prevalence increases with the severity of hypertension, reaching up to 29.1% in patients with resistant hypertension (RH). Both PA and RH are “high-risk phenotypes”, associated with increased cardiovascular morbidity and mortality compared to non-PA and non-RH patients. Aldosterone excess, as occurs in PA, can contribute to the development of a RH phenotype through several mechanisms. First, inappropriate aldosterone levels with respect to the hydro-electrolytic status of the individual can cause salt retention and volume expansion by inducing sodium and water reabsorption in the kidney. Moreover, a growing body of evidence has highlighted the detrimental consequences of “non-classical” effects of aldosterone in several target tissues. Aldosterone-induced vascular remodeling, sympathetic overactivity, insulin resistance, and adipose tissue dysfunction can further contribute to the worsening of arterial hypertension and to the development of drug-resistance. In addition, the pro-oxidative, pro-fibrotic, and pro-inflammatory effects of aldosterone may aggravate end-organ damage, thereby perpetuating a vicious cycle that eventually leads to a more severe hypertensive phenotype. Finally, neither the pathophysiological mechanisms mediating aldosterone-driven blood pressure rise, nor those mediating aldosterone-driven end-organ damage, are specifically blocked by standard first-line anti-hypertensive drugs, which might further account for the drug-resistant phenotype that frequently characterizes PA patients.

## 1. Introduction

Physiologically, aldosterone is a key hormone in the regulation of blood pressure and electrolyte homeostasis [1,2]. However, it is now broadly recognized that inappropriately high aldosterone levels with respect to their physiological regulators, as occurs in primary aldosteronism (PA), have an important role in the development of arterial hypertension and drug resistant hypertension (RH) and are associated with increased cardiovascular, metabolic, and renal complications [3,4,5,6,7,8,9].

The detrimental role of aldosterone excess depends not only on its well-known effects on water and sodium reabsorption in the kidney, but also on its pro-fibrotic, pro-inflammatory, and pro-oxidative action in several target tissues [10,11]. In this review we will focus on the potential pathophysiological mechanism by which aldosterone excess can contribute to the development of RH.

## 2. Physiological Role of Aldosterone in the Regulation of Blood Pressure

In physiological conditions, aldosterone is a key regulator of blood pressure, electrolyte balance and volume status and its actions are closely integrated with that of the renin-angiotensin-aldosterone system (RAAS) [1,2]. The most important stimulus for aldosterone secretion is represented by angiotensin II [12,13,14,15]. Therefore, all factors that stimulate the secretion of renin from the juxtaglomerular cells also lead to an increased production of aldosterone from the adrenal gland; these stimuli mostly reflect changes in extracellular fluid volume and are primarily sensed at three sites [16]: (i) stretch receptors in the afferent arteriolar wall [17]; (ii) cells of the macula densa sensing NaCl delivery to the early distal tubule [18]; (iii) vascular baroreceptors regulating sympathetic neural activity [19].

In addition, aldosterone secretion is also regulated by other angiotensin-independent molecular mechanisms. The most important is represented by extracellular potassium levels [12], which directly act by altering the driving force for potassium across the plasma membrane of the cells in the zona glomerulosa of the adrenal cortex; an increase in serum potassium thus determines a membrane depolarization, which ultimately leads to increased aldosterone secretion [20,21]. On the other hand, the adrenocorticotropic hormone (ACTH), though being recognized as an aldosterone secretagogue at the molecular level, concretely plays a minor role in the overall regulation of its production [12]; in fact, though being capable to raise aldosterone secretion acutely, ACTH is not able, differently from angiotensin II and potassium, to stimulate it chronically [22]. The overall summary of the main factors regulating and influencing aldosterone secretion is reported in Figure 1.

Shifting the focus on aldosterone actions, aldosterone fulfills its function on several target tissues via activation of the mineralocorticoid receptor (MR), an intracellular transcription factor that can modulate the transcription of effector proteins as well as initiate rapid non-genomic, or extra-nuclear, events through various signaling pathways [23].

The primary effect of aldosterone is exerted on the kidney, where it increases the expression of epithelial sodium channels (ENaC) in the luminal membrane of the principal cells in the collecting tubules [24,25]; this effect is mediated by the MR, which is located in the cytosol and—upon binding with aldosterone—exerts a transcriptional control over ENaC expression and activity [26,27]. When exposed on the luminal membrane, ENaC mediates sodium reabsorption from the tubular lumen [25,27]; being sodium reabsorbed as a cation, this determines a shift of positive electrical charges from the lumen to the tubular cells, thereby creating an electrochemical gradient that enhances the excretion of cellular potassium into the lumen through specific potassium-channels [27,28,29]. In addition, though quantitatively less important, a stimulating action of aldosterone on the expression of the thiazide-sensitive Na-Cl symporter in the distal convoluted tubule has been also demonstrated [30]. Overall, these renal actions of aldosterone promote sodium retention and potassium elimination, with a consequent extracellular fluid volume expansion and blood pressure increase [24,28,31]; in healthy subjects, this represents a “physiologically desired” response that helps maintaining the hydro-electrolytic homeostasis.

Although the kidney represents the main target organ for aldosterone action, aldosterone also binds to the MR in various other tissues, inducing pleiotropic effects. In fact, the MR has been identified in a wide variety of different cell types, most notably in the cardiovascular system (endothelial cells, vascular smooth muscle cells [VSMC], and cardiomyocytes) [32,33,34,35,36], in the neural system (hippocampal and hypothalamic neural cells) [37,38] and in the adipose tissue (adipocytes and pre-adipocytes) [39]. The available evidence suggests that the activation of MR in at least some of these tissues concurs with renal actions in regulating blood pressure levels, both through direct and indirect mechanisms [35,40]. This is even more evident in pathological conditions of aldosterone excess, as will be discussed in the following sections.

## 3. Pathological Consequences of Aldosterone Excess

As discussed previously, aldosterone plays a central role in the physiological regulation of blood pressure, stimulating several responses in various target cells in order to maintain hydro-electrolytic homeostasis. High aldosterone levels are not pathological “per se”, and do not necessarily determine arterial hypertension, as long as they are appropriate to the sodium/volume status of the individual. An outstanding example of this is provided by Yanomamo Indians, a tribe inhabiting the tropical rainforest between northern Brazil and southern Venezuela, which was renowned for living on a diet with an extremely low content of salt (approximately 10 mmol per day) [41]. These individuals were characterized by extremely high aldosterone levels (mean serum aldosterone 1475 ng/L) [42], well above the reference range used in Western countries. However, these levels were still appropriate for the specific homeostatic regulation of these subjects and, in fact, did not determine any pathological rise in arterial blood pressure [42]. Therefore, the key factor for the development of hypertension in conditions of aldosterone excess is that aldosterone levels are not only high, but inappropriately elevated with respect to the electrolyte and volume status of the individual and non-modulable by homeostatic stimuli, as occurs in PA. In this context, aldosterone excess determines several detrimental effects in its target tissues, ultimately leading to various pathological consequences [43,44,45,46,47,48].

In most cases, PA is sporadic, and the underlying cause is represented either by an aldosterone-producing adenoma (APA) or by bilateral adrenal hyperplasia (BAH) [43,44,45,46]. The distinction between bilateral and unilateral forms of PA is essential for the choice of the most appropriate therapeutic intervention (i.e., adrenalectomy in unilateral disease versus treatment with MR antagonists [MRA] in bilateral disease) [43,44,45,46,47,48]. The gold standard for the subtype diagnosis of PA is represented by adrenal vein sampling (AVS) [43,44,45,46,47,48,49]; being a technically challenging procedure, AVS is currently performed only in specialized centers, but the recent development of scores that combine clinical/biochemical variables and AVS results may help in the subtype diagnosis of PA even in the presence of a suboptimal procedure, leading to a more widespread use of AVS [50,51,52].

Once thought to be rare, PA is now recognized as the most common cause of secondary hypertension; its prevalence increases with the severity of hypertension and can be estimated approximately as 5% in primary care practice [53,54,55,56], 10% in referral centers [55,56], and up to 29.1% among patients with RH [3,57,58]. The clinical relevance of PA, however, goes far beyond its prevalence; in fact, aldosterone excess determines an increased risk of cardiovascular events and target organ damage that is independent from the degree of blood pressure elevation. Patients with PA suffer from an increased risk of atrial fibrillation, heart failure, coronary artery disease, stroke, and cardiovascular mortality, compared to matched essential hypertensives with similar blood pressure levels [3,4,5,6,7,8,9]. Furthermore, the availability of specific/targeted treatments makes the correct identification of PA patients of even greater importance; in fact, treating PA either by unilateral adrenalectomy or by MRA determines not only the cure or the improvement of hypertension [43,45,59], but also the reduction of the adjunctive cardiovascular risk to which PA is associated [60,61,62]. Therefore, since spontaneous remission of PA in RH is unlikely [63], an accurate and timely diagnosis is essential to ensure to all PA patients a specific surgical or medical treatment, whose benefits are by far greater than those of a non-specific anti-hypertensive management both in terms of blood pressure control and of cardiovascular disease prevention [43,45,59,60,61,62].

The most important mechanisms that lead to an abnormal increase in blood pressure levels in patients with PA are those exerted on the kidney. At a renal level, in fact, aldosterone excess determines an inappropriately high distal tubular resorption of sodium, which prevails on the actual sodium/volume status and leads to extracellular volume expansion [24,25,26,27]. However, in addition to these effects, there is a growing body of evidence showing that aldosterone excess may have deleterious consequences on many target tissues, by exerting a pro-fibrotic, pro-inflammatory and pro-oxidative action [10,11]. These “non-classical” effects of aldosterone, that will be detailed in the following sections, can help explain the excess cardiovascular, renal, and metabolic complications observed in patients with PA with respect to patients with essential hypertension.

## 4. Clinical Evidence on the Association between PA and RH

RH is defined as uncontrolled blood pressure despite appropriate lifestyle measures and treatment with ≥3 anti-hypertensive agents of different classes at maximal or maximally tolerated doses, including a diuretic and, typically, a calcium-channel blocker (CCB) and a blocker of the renin-angiotensin system (i.e., an angiotensin-converting enzyme inhibitor [ACE-i] or an angiotensin receptor blocker [ARB]) [64]. RH is a high-risk phenotype, associated to increased cardiovascular morbidity and mortality compared to non-RH patients; various studies demonstrated that subjects with RH are at increased cardiovascular risk independently of blood pressure levels, and that the high cardiovascular disease burden seen in RH patients persists even after effective blood pressure control is achieved [65,66,67,68,69].

PA and RH frequently coexist. In a recent study by our group, which enrolled 110 consecutive patients with true RH, PA was diagnosed in 32 cases (29.1%) [3]. A previous study by Calhoun et al. in 88 patients with RH reported a slightly lower prevalence of PA (20%) [57]; similarly, in a more recent study by Brown et al. in 408 patients with RH, PA was diagnosed in 22% of cases [70]. Therefore, given the high prevalence of PA in patients with RH, current guidelines recommend screening for PA in all patients with RH [43]. More importantly, lesser degrees of aldosterone excess that do not fulfill strict criteria for PA diagnosis may be encountered in patients with RH [71]. In a study by Gaddam et al. in 279 patients with RH compared to 53 controls (with normotension or hypertension controlled by ≤2 antihypertensive medications), the authors showed higher plasma aldosterone concentrations and lower plasma renin activity in the RH group [72]. Vice versa, in the AVIS-2-RH study, which enrolled 1625 hypertensive patients with a defined diagnosis of PA, the prevalence of RH was 20% [73]. In the same study, unilateral adrenalectomy resolved resistance to antihypertensive treatment in all RH patients who underwent surgery [73].

The high prevalence of PA in patients with RH strongly suggests a role for aldosterone excess in the development of resistance to antihypertensive medications. Moreover, it is now well established that add-on treatment with a MRA in patients with RH, even in the absence of a clear diagnosis of PA, is effective in lowering blood pressure, as highlighted in the PATHWAY-2 study, the first randomized controlled trial comparing different antihypertensive treatments in patients with RH [74]. In this study, spironolactone was the most effective add-on drug in patients whose blood pressure was not controlled by an ACE-i/ARB plus a CCB and a thiazide diuretic [74]. Moreover, the response to MRA treatment had a clear inverse relation with plasma renin, being spironolactone especially effective at lower plasma renin levels, and yet the most effective drug throughout the range of plasma renin with respect to bisoprolol and doxazosin [74]. Therefore, these findings suggest a potential role for aldosterone excess and MR activation in the development of RH even in the absence of a defined diagnosis of PA.

## 5. Pathophysiological Role of Aldosterone Excess in the Development of a RH Phenotype

Aldosterone excess can contribute to the development of severe hypertension and resistance to antihypertensive treatment through several mechanisms.

### 5.1. Salt Retention and Volume Expansion

The most straightforward mechanism by which aldosterone excess can contribute to the development of RH is the chronic expansion of blood volume and enhanced salt retention [24,25,26,27,45,47,48]. At a renal level, in fact, aldosterone excess determines an inappropriately high distal tubular resorption of sodium, which prevails on the actual sodium/volume status and leads to extracellular volume expansion [24,25,26,27].

The relevance of body sodium content and water retention as a determinant of blood pressure levels has been demonstrated by clinical studies, in which lifestyle interventions decreasing daily salt intake have been shown to significantly improve blood pressure control, even without changes in prescribed medications [75]. Sodium retention and volume expansion have been shown to play a key role especially in the pathophysiology of RH: pathophysiological analyses of endocrine and hemodynamic features of RH have shown that this condition commonly presents with the features of a chronic volume and salt-retaining state, even in subjects without a biochemical diagnosis of PA [76]. Atrial natriuretic peptide (ANP) and brain natriuretic peptide (BNP), that are produced in the cardiac atria and ventricles in response to volume or pressure overload, have been shown to be higher in patients with RH compared to controls, suggesting that increased intravascular volume is a common characteristic of RH [72]. The PATHWAY-2 mechanistic substudies demonstrated that the blood pressure response to spironolactone in RH is associated with elimination of thoracic volume excess rather than vasodilatation, indicating that RH is attributable in large part to excess fluid retention mediated by aldosterone excess [77]. These results are consistent with the findings of prior studies in patients with RH, which demonstrated an increased intravascular fluid retention, evidenced by high natriuretic peptide levels, as well as an increase in intracardiac volumes measured by magnetic resonance imaging in this population [71]. Therefore, as we have already mentioned, correcting the volume expansion by MRA seems to be the best option to overcome treatment resistance in patients with uncontrolled blood pressure during a 3-drug combination regimen [77,78].

In conclusion, inappropriate fluid retention secondary to high dietary sodium intake and aldosterone excess is an important mediator of antihypertensive treatment resistance that is best overcome by the natriuretic and diuretic effects of spironolactone in patients with RH [71]. Coherently, current guidelines recommend MRA as the preferred drug class that should be added as a fourth medication in RH patients [64,79].

### 5.2. Oxidative Stress, Inflammation, Endothelial Dysfunction, and Fibrosis

As we have already mentioned, aldosterone excess has been shown to have detrimental effects on many target organs that go beyond its action on water and sodium reabsorption and that can also contribute to the development of a RH phenotype. The deleterious effects of aldosterone excess at a vascular level (i.e., oxidative stress, inflammation, endothelial dysfunction, fibrosis, vascular remodeling, and increased arterial stiffness) [10,80,81,82,83,84] and at a renal level (i.e., hypertrophy/hyperplasia of distal tubule cells, oxidative stress, vascular and tubular inflammation and fibrosis) [85,86,87] likely determine and promote a vicious cycle, that ultimately leads to a more severe hypertensive phenotype and to a quicker progression of target organ damage.

One way by which aldosterone promotes inflammation is by increasing the generation of reactive oxygen species (ROS), which in turn activate pro-inflammatory transcription factors such as nuclear factor kappa-light-chain-enhancer of activated B cells (NFkB). Aldosterone can increase oxidative stress through several mechanisms. First of all, it can induce the activation of nicotinamide adenine dinucleotide phosphate (NADPH) oxidase in several organs (blood vessels, heart, kidney) [88]. Moreover, it can decrease the vascular expression of glucose-6-phosphate dehydrogenase, thereby reducing the production of NADPH; NADPH is in turn required for the generation of reduced glutathione, which constitutes an important defense against oxidative damage [89]. In addition, aldosterone can determine the uncoupling of nitric oxide synthase (NOS), thereby leading to the production of superoxide instead of nitric oxide (NO) and can also stimulate mitochondrial generation of ROS [88]. Finally, aldosterone has been shown to promote inflammation directly via the MR-mediated activation of serum/glucocorticoid regulated kinase 1 (SGK1) [90].

Aldosterone-induced oxidative stress and inflammation can promote endothelial dysfunction and increase the expression of pro-fibrotic molecules, such as transforming growth factor-β1 (TGF-β1), connective tissue growth factor (CTGF), placental growth factor (PGF), endothelin 1 (ET1), osteopontin and galectin-3, eventually leading to fibrosis in the heart, kidney and vasculature [88]. The mechanisms underlying the pro-oxidant and pro-inflammatory effects of aldosterone are summarized in Figure 2. 

Aldosterone excess can cause endothelial dysfunction by determining an imbalance between vasoconstrictors and vasodilators. First of all, aldosterone impairs NO formation via a MR-dependent pathway by causing a reduction of cofactor BH4, by determining the dephosphorylation of endothelial NOS, and by increasing the production of ROS and impairing ROS scavenging capacity, thus reducing the bioavailability of NO [91,92]. Moreover, aldosterone has been shown to increase the production of vasoconstrictors, such as prostacyclin, thromboxane A2 and ET1 by inducing cyclooxygenase-2 activation in rat models [93,94,95].

The MR is expressed not only in endothelial cells, but also in VSMCs. In the presence of endothelial damage, which can be cause by mechanical injury or by hypertension, dyslipidemia, diabetes and smoking, VSMC are capable of proliferating and producing extracellular matrix thereby contributing to vascular fibrosis and vessel wall thickening and stiffening, a process known as vascular remodeling. Studies on VSMC-MR knockout mice have shown that VSMC-MR is essential for aldosterone-mediated vascular fibrosis and stiffening, since in the absence of VSMC-MR aldosterone does not mediate its detrimental effects on the vasculature [96]. In animal models, aldosterone has been shown to induce fibrosis of the arterial wall via different molecular pathways. In a study by Park et al. in aldosterone-infused rats, aortic collagen and media cross sectional area were significantly increased, probably via an ET1-mediated mechanism [97]. In another study by Harvey et al. in stroke-prone spontaneously hypertensive rats, authors demonstrated that aldosterone can induce vascular fibrosis via NADPH oxidase 1-mediated activation of p66Shc, an important redox-sensitive signaling molecule that has been implicated in cardiovascular aging, vascular injury and renal dysfunction [98]. Other studies have suggested a key role of galectin-3, a beta-galactoside-binding lectin, in aldosterone-mediated vascular fibrosis: in a study by Calvier et al. in a rat model, galectin-3 has been shown to mediate aldosterone-induced increase of collagen type 1 in the aortic wall [99]. Several studies have also suggested a role for Rho-kinase signaling, a pathway that regulates a wide range of SMC functions, including contraction, migration, proliferation, and apoptosis [100,101,102]. Additional factors that have been implicated in aldosterone-induced vascular fibrosis are CTGF and TGF-β1, which promote fibrosis by stimulating cellular transformation to fibroblasts, by increasing the synthesis of matrix proteins and integrins, and by decreasing the production of MMPs, osteopontin, an extracellular matrix protein that promotes adhesion of inflammatory cells, and PGF, a member of the vascular endothelial growth factor family that induces vascular cell proliferation and extracellular matrix deposition [88].

At a vascular level, aldosterone-mediated oxidative stress, inflammation, endothelial dysfunction, and fibrosis [10] lead to increased arterial stiffness and peripheral vascular resistance, factors that have both been implicated in the pathogenesis of RH and in the progression of target organ damage [103]. In clinical studies, patients with PA are characterized by increased carotid intima-media thickness, higher arterial stiffness as measured by pulse wave velocity, and altered flow mediated dilation [83,84,104,105,106,107] with respect to patients with essential hypertension.

Aldosterone pro-oxidative, pro-inflammatory, and pro-fibrotic effects have also been demonstrated in the kidney. In animal models, aldosterone excess has been shown to induce sodium-dependent hypertrophy and hyperplasia of distal tubule cells [85,86], as well as vascular and tubular inflammation and fibrosis [87]. Moreover, patients with PA are characterized by higher levels of albuminuria compared to essential hypertensives [108,109,110], which represents a marker of renal damage and renal damage progression. Animal models suggest that aldosterone-induced kidney injury is mediated by NADPH-oxidase dependent ROS production, that lead to podocyte damage and transcription of proinflammatory and pro-fibrotic factor via NFkB and Rho-kinase, as well as by EGFR-mediated PI3K-AKT and MAPK activation, that induce mesangial cell proliferation [111].

The relationship between CKD and RH is bilateral: on the one hand, uncontrolled blood pressure leads to the development of renal complications; on the other hand, CKD is recognized as a frequent cause of RH in the general hypertensive population. RH is a common finding among patients with CKD: in a study by De Nicola et al., evaluating 436 hypertensive CKD patients, true RH had a prevalence of 22.9% [112]. Several factors may contribute to antihypertensive drug resistance in CKD: impaired ability to excrete salt intake, increased production of inflammatory molecules and ROS, endothelial dysfunction and increased sympathetic nervous activity [113].

### 5.3. Sympathetic Nervous System Overactivity

Marked neuroadrenergic activation and baroreflex dysfunction have been shown in patients with true RH as compared with apparent RH and essential hypertension [114,115]. At a neural level, there is evidence that aldosterone can increase sympathetic nervous system (SNS) activity [82,116] and impair baroreflex response [117,118], probably via a non-genomic mechanism, overall determining an increased sympathetic outflow and a blood pressure rise [119]. A cross-sectional comparison analysis between patients with PA, essential hypertensives and normotensive controls demonstrated that muscle sympathetic nerve activity (SNA) measured with intraneural microelectrodes was significantly higher in the PA than the normotensive group. Moreover, after adrenalectomy for unilateral form of PA, SNA decreased significantly. These data provide evidence that unilateral forms of PA are accompanied by reversible sympathetic overactivity, which may contribute to the accelerated hypertensive target organ disease in this condition [120].

Several studies in literature have suggested that the effect of aldosterone on SNS activity may be also dependent on MR-mediated effects of aldosterone in the central nervous system. In a study by Wray et al., the authors reported a significant reduction in SNS activity after six months of therapy with MRAs, which was obtained without a change in end-organ alpha-adrenergic responsiveness, implicating a central mechanism for the reduced autonomic activity [121]. Expression of MR has been demonstrated in the paraventricular nucleus (PVN), a hypothalamic nucleus involved in the regulation of sympathetic drive; MR-blockade has been shown to decrease NADPH oxidase activity and superoxide generation in the PVN of rats with heart failure, with a concomitant reduction of chronic excitation of neurons in the PVN and plasma NE levels [122,123].

### 5.4. Adipose Tissue Dysfunction, Insulin Resistance, and Obesity

Aldosterone excess might promote and sustain a rise in blood pressure also through some additional indirect pathways involving adipose tissue dysfunction. In fact, there is some evidence supporting a possible role of PA in the pathogenesis of insulin-resistance and metabolic syndrome, which are per se well-known risk factors and substrates for the development and sustenance of hypertension [124]. The prevalence of metabolic syndrome has been shown to be higher among patients with PA than in those with essential hypertension [125]; moreover, some studies have reported an association between aldosterone levels and BMI [126,127]. From a molecular point of view, this might be a consequence of the action exerted by aldosterone on adipose tissue function: aldosterone has been shown to reduce the release of insulin-sensitizing mediators by the adipocytes and pre-adipocytes by inducing pro-inflammatory effects in the adipose tissue and altering adipokine expression [128,129], thus promoting insulin-resistance both in animal models [130,131] and in humans [132]. Aldosterone has also shown inhibitory effects on insulin signaling and insulin-stimulated glucose uptake via glut-4 in adipocytes, skeletal muscle, and vascular smooth muscle cells [128].

Aldosterone excess and obesity also have a causative role in the development of obstructive sleep apnea (OSA), a condition that appears to be strongly related to RH: in a study on 41 patients with RH, Logan et al. diagnosed OSA in 83% of cases; moreover, the severity of OSA was correlated with the presence of RH [133]. Further, the more severe the sleep apnea, the more likely the resistance to antihypertensive therapy [134]. The proposed mechanisms by which OSA can contribute to the development of a RH phenotype are vascular stiffening due to repeated intermittent arousals, increased levels of circulating vasoconstrictors such as norepinephrine and endothelin and sympathetic activation [123,135]. The link between aldosterone excess and OSA is bilateral: on the one hand, the SRAA appears to be overactivated in patients with OSA and several studies have demonstrated that optimal treatment with continuous positive air pressure ventilation significantly decreases aldosterone levels; on the other hand, the volume overload associated with aldosterone excess may have a role in the pathogenesis of OSA, in which a nocturnal rostral fluid shift has been demonstrated that in turn leads to tissue edema and airflow obstruction [136].

### 5.5. Lower Efficacy of Anti-Hypertensive Drugs

The specific pathophysiology of hypertension in patients with PA also influences the efficacy of standard first-line anti-hypertensive medications. In fact, the pathophysiological mechanisms underlying the development of hypertension in patients with PA are persistent and sustained even during treatment with a conventional 3-drug combination regimen (which typically includes a diuretic, a CCB, and either an ACEi or an ARB); the driving mechanisms of sodium retention and volume expansion that characterize the pathophysiology of hypertension in PA are not specifically hampered by standard first-line anti-hypertensive drugs, and this might decrease their efficacy in blood pressure control [43,46,71,137]. Moreover, with respect to ACEi and ARB, their efficacy has been shown to be significantly correlated to plasma renin levels, with a lower blood pressure reduction in low-renin states, irrespectively to the presence of PA; this could further explain the relatively lower efficacy of these drug classes in PA patients [77].

In addition, non-specific anti-hypertensive treatment has been demonstrated to be insufficient to provide optimal organ protection in patients with PA; the available evidence, in fact, highlights that the benefit of a specific surgical or medical treatment of PA is significantly higher than the one of non-specific anti-hypertensive medications, both in terms of clinical outcomes and end-organ damage [58,59,138,139]. Therefore, treating PA patients with anti-hypertensive medications other than MR antagonists does not prevent the blood-pressure-independent cardiac and vascular damage that is determined by inappropriate aldosterone excess [58,59,138,139]. Ultimately, treatment resistance in PA patients is not limited to the lower efficacy of standard anti-hypertensive medications in lowering blood pressure levels per se, but also probably extends to the inability of these drugs to stop the progression of blood-pressure-independent cardiovascular end-organ damage, which ultimately might determine a more severe hypertension phenotype in the medium-to-long term.

The overall summary of the main pathophysiological mechanisms by which PA may determine a resistant-hypertensive phenotype is reported in Figure 3.

## 6. Conclusions

In this review we have summarized the current evidence regarding the potential pathophysiological link between aldosterone excess, as occurs in PA, and RH.

The most important mechanisms leading to the development and maintenance of arterial hypertension and RH in patients with PA are those exerted on the kidney, with increased sodium retention and volume expansion. These mechanisms are believed to play a key role in the pathophysiology of RH even in non-PA patients. Moreover, there is a growing body of evidence showing that the excessive activation of MR in other tissues may also play a role, mostly through vascular remodeling, increased sympathetic outflow, and additional indirect pathways involving insulin-resistance and adipose tissue dysfunction. The pro-oxidative, pro-fibrotic and pro-inflammatory effects of aldosterone may also contribute to the worsening of end-organ damage, thereby perpetuating a vicious cycle that eventually leads to a more severe hypertensive phenotype. Finally, neither the pathophysiological mechanisms mediating aldosterone-driven blood pressure rise, nor those mediating aldosterone-driven end-organ damage, are specifically blocked by standard first-line anti-hypertensive drugs, which might further account for the drug-resistant phenotype that frequently characterizes PA patients.

Both PA and RH are “high-risk phenotypes”, associated with increased cardiovascular morbidity and mortality compared to non-PA and non-RH patients. Given the high prevalence of PA in patients with RH, an accurate screening of PA in this population is essential, to ensure to all PA patients a timely diagnosis and a specific surgical or medical treatment, whose benefits are by far greater than those of non-specific anti-hypertensive medications, both in terms of blood pressure control and of cardiovascular damage prevention. Moreover, acknowledging the role of aldosterone excess in the development of RH even in the absence of a clear diagnosis of PA has important implications for the choice of the most effective add-on antihypertensive medications.

## Figures and Tables

**Figure 1 ijms-23-04803-f001:**
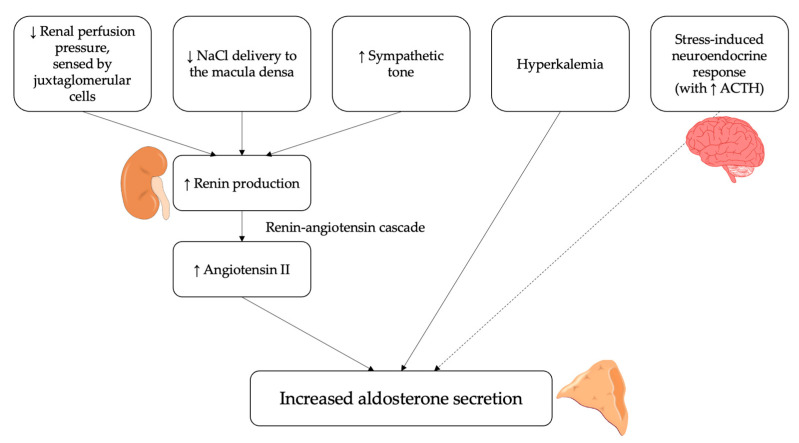
Physiological regulators of aldosterone secretion. Several stimuli can induce the release of aldosterone by the adrenal cortex via the renin-angiotensin system. These include a reduction in renal perfusion pressure, sensed by juxtaglomerular cells, a decrease in NaCl delivery to the macula densa and an increase in sympathetic activity. Moreover, hyperkalemia can directly stimulate aldosterone secretion from adrenal cortex. Finally, the adrenocorticotropic hormone (ACTH) can act as an aldosterone secretagogue, although it plays a minor role in its regulation.

**Figure 2 ijms-23-04803-f002:**
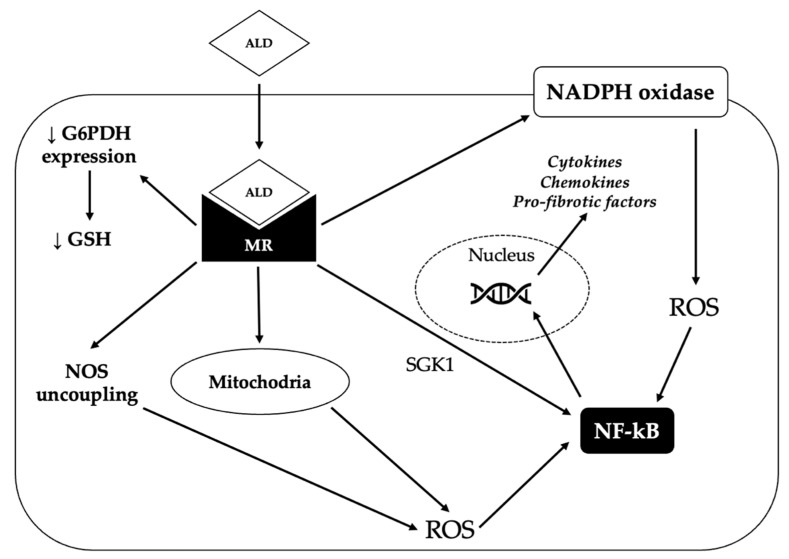
Mechanisms underlying the pro-oxidant and pro-fibrotic effect of aldosterone. By binding to the mineralocorticoid receptor (MR), aldosterone (ALD) can induce the production of reactive oxygen species (ROS) through several mechanisms. First of all, aldosterone can promote the generation of ROS by activating the nicotinamide adenine dinucleotide phosphate (NADPH) oxidase. Moreover, MR activation increases the production of ROS by the mitochondria and can also determine the uncoupling of nitric oxide synthase (NOS), thereby leading to the production of superoxide instead of nitric oxide. Finally, aldosterone can reduce glucose-6-phosphate dehydrogenase (G6PDH) expression, thereby impairing the generation of reduced glutathione (GSH), one of the main defensive mechanisms of the cell against oxidative stress. Eventually, increased ROS production triggers the activation of nuclear factor kappa-light-chain-enhancer of activated B cells (NF-kB), that in turn induces the transcription of pro-inflammatory and pro-fibrotic genes. Activated MR can also directly stimulate NF-kB, probably via serum/glucocorticoid regulated kinase 1 (SGK1).

**Figure 3 ijms-23-04803-f003:**
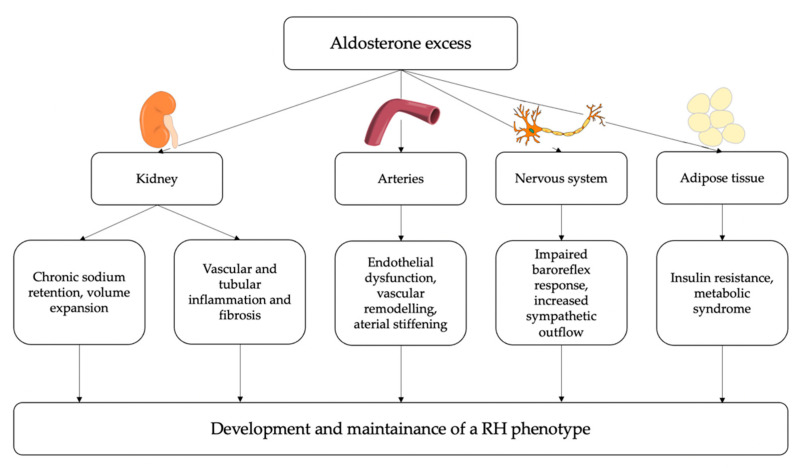
Potential pathophysiological mechanisms linking aldosterone excess and resistant hypertension (RH). At a renal level, aldosterone induces chronic sodium and water retention, leading to intravascular volume expansion; moreover, aldosterone can contribute to vascular and tubular inflammation and fibrosis, thereby impairing kidney function. At a vascular level, aldosterone induces endothelial dysfunction, oxidative stress, inflammation and fibrosis thereby leading to arterial stiffening. Additionally, aldosterone excess can directly influence sympathetic activity and can also have a role in the development of insulin resistance and metabolic syndrome. Eventually, these detrimental effects of aldosterone excess on several target tissues can contribute to the development and maintenance of resistant hypertension.

## Data Availability

Not applicable.

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
