# Peer review of "Primary Aldosteronism and Resistant Hypertension: A Pathophysiological Insight"

_ijms, 2022, doi:10.3390/ijms23094803_

Round 1

Reviewer 1 Report

The review “Primary aldosteronism and resistant hypertension: a patho-2 physiological insight” by Bioletto, F., et al., describes about the key mechanisms that are responsible for the development and maintenance of arterial hypertension and resistant hypertension (RH) not only in patients with primary aldosteronism but also in subjects with non-PA. According to authors the pro-oxidative, pro-fibrotic and pro-inflammatory effects of aldosterone are primarily contributing to the worsening of end-organ damage that leads to a more severe hypertensive phenotype. Authors have also mentioned that the pathophysiological mechanisms mediating aldosterone-driven blood pressure rise, nor those mediating aldosterone-driven end-organ damage could be blocked by standard anti-hypertensive drugs, which might be accounting for the drug-resistant phenotype of PA patients. Authors are of opinion that (a) both PA and RH are high-risk phenotypes indicative of elevated cardiovascular morbidity and mortality compared to non-PA and non-RH patients; and (b) Accurate screening of PA in patients with RH is essential, as it helps not only in timely diagnosis but also in providing a specific surgical or medical treatment.

Even though the review is well written and interesting, in its current form, it is too wordy and requires the addition of appropriate figures and tables (Suggestions provided in the comments section).

Comments

  1. Provide detailed figure legends
  2. Provide figures for each of the effects of aldosterone-regulated mechanisms. Providing these figures will help the reader to better understand the molecular mechanisms that lead to the development of pro-inflammatory effects etc
  3. Many reviews, similar to the one presented in this article, have been published previously.

Int J Hypertens. 2011; 2011: 837817.

Published online 2011 Jan 20. doi: 10.4061/2011/837817

Aldosteronism and Resistant Hypertension

Maria Czarina Acelajado1,* and David A. Calhoun1, 2

Physiological Reviews, 2016, 96(4)

Primary Aldosteronism: Changing Definitions and New Concepts of Physiology and Pathophysiology Both Inside and Outside the Kidney

Michael Stowasser and Richard D. Gordon

Journal of Internal Medicine

Diagnosis and treatment of primary aldosteronism: practical clinical perspectives

  1. F. Young Jr

First published: 25 September 2018

Therefore, it would be better if authors could provide additional information from recent research findings and how add their own thoughts, which helps in designing better studies

  1. Authors would have discussed about the current treatment options their merits and demerits; and possible ways to move forward in this area of research

Author Response

Reviewer 1

The review “Primary aldosteronism and resistant hypertension: a patho-2 physiological insight” by Bioletto, F., et al., describes about the key mechanisms that are responsible for the development and maintenance of arterial hypertension and resistant hypertension (RH) not only in patients with primary aldosteronism but also in subjects with non-PA. According to authors the pro-oxidative, pro-fibrotic and pro-inflammatory effects of aldosterone are primarily contributing to the worsening of end-organ damage that leads to a more severe hypertensive phenotype. Authors have also mentioned that the pathophysiological mechanisms mediating aldosterone-driven blood pressure rise, nor those mediating aldosterone-driven end-organ damage could be blocked by standard anti-hypertensive drugs, which might be accounting for the drug-resistant phenotype of PA patients. Authors are of opinion that (a) both PA and RH are high-risk phenotypes indicative of elevated cardiovascular morbidity and mortality compared to non-PA and non-RH patients; and (b) Accurate screening of PA in patients with RH is essential, as it helps not only in timely diagnosis but also in providing a specific surgical or medical treatment.

Even though the review is well written and interesting, in its current form, it is too wordy and requires the addition of appropriate figures and tables (Suggestions provided in the comments section).

Comments

  1. Provide detailed figure legends
  2. Provide figures for each of the effects of aldosterone-regulated mechanisms. Providing these figures will help the reader to better understand the molecular mechanisms that lead to the development of pro-inflammatory effects etc

Thank you very much for the comments and suggestions. We agree with the reviewer, providing more detailed figure legends and adding a figure on molecular mechanisms that lead to aldosterone mediated damage. All the changes made on the manuscript are highlighted in yellow.

  1. Many reviews, similar to the one presented in this article, have been published previously.

Int J Hypertens. 2011; 2011: 837817.

Published online 2011 Jan 20. doi: 10.4061/2011/837817

Aldosteronism and Resistant Hypertension

Maria Czarina Acelajado1,* and David A. Calhoun1, 2

Physiological Reviews, 2016, 96(4)

Primary Aldosteronism: Changing Definitions and New Concepts of Physiology and Pathophysiology Both Inside and Outside the Kidney

Michael Stowasser and Richard D. Gordon

Journal of Internal Medicine

Diagnosis and treatment of primary aldosteronism: practical clinical perspectives

  1. Young Jr

First published: 25 September 2018

Therefore, it would be better if authors could provide additional information from recent research findings and how add their own thoughts, which helps in designing better studies

Many thanks for the suggestions and for the eminent papers considered. Prior to the writing of this paper, an extensive literature research has been made in order to offer an updated revision about the mechanisms that lead patients with autonomous aldosterone secretion to treatment resistant hypertension and cardiometabolic damage. Only the first review (Int J Hypertens. 2011; 2011: 837817), which has benn published in 2011, is partially focused on this topic, but not the other two papers (Physiological Reviews, 2016, 96(4); J Intern Med. 2019 Feb;285(2):126-148). We think that our review can help the readers, giving an updated state of the art of the pathophysiological insight of resistant hypertension in the context of primary aldosteronism, summarizing all the relevant evidence of the last years.

  1. Authors would have discussed about the current treatment options their merits and demerits; and possible ways to move forward in this area of research

Thank you very much for the suggestions. It would be very interesting to discuss about the current treatment options for resistant hypertension and primary aldosteronism, but this topic needs another specific paper and unfortunately, this is not the focus of our review. The scope of this review was to summarize in an updated article all the pathophysiological mechanisms that lead patients with primary aldosteronism to the development of treatment resistant hypertension.

Reviewer 2 Report

This is a crisp, updated review on primary aldosteronism (PA) and resistant hypertension that is interesting and enjoyable to read. I congratulate the authors on their work.

Just a few minor comments, hoping they will be helpful.

- Section 3 could be clearer, in my opinion. The authors begin by talking about PA, how it is diagnosed, its clinical consequences, and treatment implications. Then they mention the Yanomamo Indians to explain that it is not the absolute increase in aldosterone levels per se that has clinical repercussions, but its relationship to the individual's ideo-electrolyte state. Finally, the authors mention the physio-pathological mechanisms of these clinical consequences, which are discussed more extensively in the following sections. Not critical to me, but I suggest changing the order of the arguments as follows: explain that increased aldosterone levels are not pathological per se but in relation to the ideo-electrolytic status of the individual (with the example of the Yanomamo Indians), then talk about PA including its clinical consequences. I am not sure that I would mention here the pathophysiological aspects, since these are discussed in detail later and it sounds a bit repetitive as it stands.

- Similarly, I propose that section 5.2 be organized differently, as it is a bit chaotic and repetitive. In particular, I suggest deleting the paragraph on page 7, lines 300-303 and putting the one currently on page 6, lines 259-264 in its place. Also, I suggest following the same order of topics as given in the subheading. For example: oxidative stress and inflammation, vascular damage including endothelial dysfunction, vascular remodeling and fibrosis, and kidney damage.

- In Figure 1, I would remove the icon of potassium, which is not very informative, in my opinion.

- I recommend that authors check their manuscript for typos. Some examples below.

- In the abstract, the phrase "inappropriate aldosterone levels with respect to the hydro-electrolytic status individual can induce salt retention and volume expansion by determining sodium and water reabsorption in the kidney" (page 1, lines 15-16) is not clear to me.

- In the sentence "The relevance of body sodium content and retention as a determinant of blood pressure levels has been demonstrated by other clinical studies, in which lifestyle interventions decreasing daily salt intake have been demonstrated to significantly improve blood pressure control, even without changes in prescribed medications" (page 5, lines 204-206), the verb "to be demonstrated" is repetitive.

- Page 5, lines 217-218: please correct "indicating that they indicate that".

- Page 5, lines 219-222: the sentence " These results are consistent with the findings of prior studies of increased intravascular fluid retention evidenced by high natriuretic peptide levels" is unclear to me. I recommend rephrasing it.

Author Response

Reviewer 2

This is a crisp, updated review on primary aldosteronism (PA) and resistant hypertension that is interesting and enjoyable to read. I congratulate the authors on their work.

Thank you very much for the appreciations.

Just a few minor comments, hoping they will be helpful.

- Section 3 could be clearer, in my opinion. The authors begin by talking about PA, how it is diagnosed, its clinical consequences, and treatment implications. Then they mention the Yanomamo Indians to explain that it is not the absolute increase in aldosterone levels per se that has clinical repercussions, but its relationship to the individual's ideo-electrolyte state. Finally, the authors mention the physio-pathological mechanisms of these clinical consequences, which are discussed more extensively in the following sections. Not critical to me, but I suggest changing the order of the arguments as follows: explain that increased aldosterone levels are not pathological per se but in relation to the ideo-electrolytic status of the individual (with the example of the Yanomamo Indians), then talk about PA including its clinical consequences. I am not sure that I would mention here the pathophysiological aspects, since these are discussed in detail later and it sounds a bit repetitive as it stands.

- Similarly, I propose that section 5.2 be organized differently, as it is a bit chaotic and repetitive. In particular, I suggest deleting the paragraph on page 7, lines 300-303 and putting the one currently on page 6, lines 259-264 in its place. Also, I suggest following the same order of topics as given in the subheading. For example: oxidative stress and inflammation, vascular damage including endothelial dysfunction, vascular remodeling and fibrosis, and kidney damage.

- In Figure 1, I would remove the icon of potassium, which is not very informative, in my opinion.

- I recommend that authors check their manuscript for typos. Some examples below.

- In the abstract, the phrase "inappropriate aldosterone levels with respect to the hydro-electrolytic status individual can induce salt retention and volume expansion by determining sodium and water reabsorption in the kidney" (page 1, lines 15-16) is not clear to me.

- In the sentence "The relevance of body sodium content and retention as a determinant of blood pressure levels has been demonstrated by other clinical studies, in which lifestyle interventions decreasing daily salt intake have been demonstrated to significantly improve blood pressure control, even without changes in prescribed medications" (page 5, lines 204-206), the verb "to be demonstrated" is repetitive.

- Page 5, lines 217-218: please correct "indicating that they indicate that".

- Page 5, lines 219-222: the sentence "These results are consistent with the findings of prior studies of increased intravascular fluid retention evidenced by high natriuretic peptide levels" is unclear to me. I recommend rephrasing it.

Thank you for the useful suggestions. We tried to make clearer the section 3 and to reorganize the section 5.2. We removed the icon of potassium and checked the manuscript for typos. All the changes made on the manuscript are highlighted in yellow.

Reviewer 3 Report

The work is done carefully and well written. It contains interesting information about the role of aldosterone excess in the development of resistant hypertension.  

There are errors in editing the text.

Author Response

Reviewer 3

The work is done carefully and well written. It contains interesting information about the role of aldosterone excess in the development of resistant hypertension.

There are errors in editing the text.

Thank you very much for the appreciations. We corrected the errors in editing the text.

Reviewer 4 Report

Bioletto, et al. reviewed pathophysiological mechanisms of resistant hypertension by aldosterone excess. This review is well-written and comprehensive. I just notice a typing error.

  1. Page 5, Line 217. The word “indicate” is redundant.

Author Response

Reviewer 4

Bioletto, et al. reviewed pathophysiological mechanisms of resistant hypertension by aldosterone excess. This review is well-written and comprehensive. I just notice a typing error.

Page 5, Line 217. The word “indicate” is redundant.

Thank you very much for the appreciations. We correct the redundant word.

Round 2

Reviewer 1 Report

The legends and additional details provided in the manuscript are appropriate and satisfactory